# Social factors associated with reversing frailty progression in community-dwelling late-stage elderly people: An observational study

Katsuhiko Takatori[1,2]*, Daisuke Matsumoto[1,2]

1 Department of Physical Therapy, Faculty of Health Science, Kio University, Koryo, Kitakatsuragi-gun, Nara, Japan, 2 Health Promotion Center, Kio University, Koryo, Kitakatsuragi-gun, Nara, Japan

☯ These authors contributed equally to this work.
* k.takatori@kio.ac.jp

**Data Availability Statement:** Data underlying the study can be found here: https://doi.org/10.6084/m9.figshare.12793817.v1.

## Abstract

Frailty is considered to be a complex concept based mainly on physical vulnerability, but also vulnerabilities in mental/psychological and social aspects. Frailty can be reversible with appropriate intervention; however, factors that are important in recovering from frailty have not been clarified. The aim of the present study was to identify factors that help an individual reverse frailty progression and characteristics of individuals that have recovered from frailty. Community-dwelling people aged ≥75 years who responded to the Kihon Checklist (KCL) were enrolled in the study. The KCL consists of 25 yes/no questions in 7 areas: daily-life related activities, motor functions, nutritional status, oral functions, homebound, cognitive functions, and depressed mood. The number of social activities, degree of trust in the community, degree of interaction with neighbors, and subjective age were also evaluated. Frailty was assessed based on the number of checked items: 0–3 for robust, 4–7 for pre-frailty, and ≥8 for frailty. A total of 5050 participants were included for statistical analysis. At the time of the baseline survey in 2016, 18.7% (n = 942) of respondents had frailty, and the follow-up survey showed that the recovery rate from frailty within 2 years (median 24 months) was 31.8% (n = 300). Multiple logistic regression analysis showed that exercise-based social participation (odds ratio [OR] 2.0, 95% confidence interval [CI] 1.2–3.4; $P$<0.01) and self-rated health (OR 1.2, CI 1.0–1.5; $P$ = 0.02) were related to reversing frailty progression. Principal component analysis indicated that the main factors constituting the first principal component (contribution rate, 18.3%) included items related to social capital, such as interaction with neighbors, trust in the community, and number of social participation activities. Our results demonstrate that exercise-based social participation and high self-rated health have associations with reversing frailty progression. Individuals that recovered from frailty are characterized by high individual-level social capital components (i.e., trust in community, interaction with neighbors, and social participation).

**Funding:** The authors received no specific funding for this work.

**Competing interests:** The authors have declared that no competing interests exist.

## Introduction

The term frailty has long been used to describe a general condition in which elderly people are more vulnerable to being unable to cope with everyday activities and acute stressors [1, 2]; however, the concept was clarified in Japan after the 2014 Japanese Geriatric Society statement [3]. The concept of frailty in this statement refers to "a state in which vulnerability to stress is increased" and "a state in which the subject is likely to fall into outcomes such as impaired life function, need for nursing care, and death".

Japan is a super-aging society, and it has been estimated that the proportion of late-stage elderly people (aged ≥75 years) who are highly dependent on medical care and at risk of requiring long-term care is likely to increase [4]. Therefore, extending the healthy life expectancy of late-stage elderly people is an urgent issue. Makizako et al. [5] prospectively investigated the relationship between physical frailty and the need for nursing care over a 2-year period. They reported that, even if adjustments were made for the effects of age and gender, the risk of needing nursing care, compared with healthy elderly people, was 2.5 times higher for those with pre-frailty and 4.7 times higher for those with frailty. An analysis of the prevalence of physical frailty based on a combined analysis of 5 regional cohort studies reported that 7.4% of subjects were frail and 48.1% were pre-frail [6]. The incidence of frailty by age was 10.0% for those aged 75–79 years, 20.4% for those aged 80–84 years, and 35.1% for those aged 85 years and over. Naturally, the late-stage elderly population is at a higher risk of developing frailty than those ≤74 years old [6].

Frailty is considered to be a complex concept based mainly on physical vulnerability, but also vulnerabilities in mental/psychological and social aspects [7]. In particular, approaches to social frailty, as well as physical frailty, have attracted attention in recent years, and community activities and participation in preventive intervention have been emphasized [8, 9]. Although there is no standardized method for frailty identification, the Cardiovascular Health Study (CHS) criteria and the Frailty Index are typical evaluation methods [10, 11]. The CHS criteria represent a diagnostic method based on the phenotype model of Fried et al. [10], and the Frailty Index is a diagnostic method based on the Accumulation of Deficits Models of Mitnitski et al. [11].

In Japan, the Kihon Checklist (KCL) of the Ministry of Health, Labor and Welfare (MHLW) is used for frailty assessment [12]. The KCL is a postal self-administered questionnaire consisting of 25 "yes" or "no" questions in 7 areas, including daily-life related activities, motor functions, nutritional status, oral functions, homebound status, cognitive functions, and depressed mood. The KCL is also included in the frailty management guidelines for the Asia-Pacific region, and its validity with respect to the CHS frailty criteria has been confirmed [13]. Assessment based on the total KCL score has been useful for screening frailty status in older adults and for predicting support/care-need certification in the long-term care insurance system [14, 15].

Interventions for frailty are classified into physical exercise programs, dietary supplements, and visits to medical professionals, but most of the effects manifest as incremental improvements in parameters such as walking speed, grip strength, and physical activity. Few studies have studied recovery from frailty as a primary outcome [16–19]. Cognitive training and educational interventions for specific groups are the only measures that have been shown to have moderate efficacy [19]. Furthermore, many of these studies focused only on physical frailty, and few studies carried out evaluations of social frailty [16–19]. In addition, while it has been clarified that factors such as age, smoking status, and disease status (no history of diabetes, stroke or chronic obstructive pulmonary disease) are related to recovery from frailty, the influence of social factors has not been clarified [20]. Although a previous study conducted in

Japan reported that walking for 30 minutes or more per day, meeting friends at least once a month, and going out every day are effective in recovering from frailty, it is not clear whether or not these factors improve baseline conditions [21].

We hypothesize that not only improvement of physical function, but also individual-level social factors such as social networking with neighbors and trust in the community play an important role in improving frailty status. Therefore, the purpose of this study was to clarify physical, psychological, and social factors that contribute to reverse frailty progression.

## Methods

### Study population

The study population consisted of community-dwelling elderly people ≥75 years old (late-stage elderly) in Ikoma City, Nara Prefecture, Japan. As a baseline survey, a postal survey was conducted by the community-based integrated care division of Ikoma city using the KCL for 12,698 late-stage elderly people in April to May 2016 who were not certified as requiring long-term care. The follow-up survey included 6,517 elderly cohorts who answered the KCL (response rate 75%) in April to May 2018 (median, 24 months). By the time of the follow-up survey, 537 people were newly certified as requiring long-term care insurance service. From the perspective of long-term care prevention, excluding those who needed new long-term care services and 930 participants who did not respond to the KCL or were missing (moving or death), 5050 people were included in the analysis. A flow chart of the participant selection process is shown in Fig 1.

The study protocol was approved by the ethics committee of Kio University (approval number: H28–57). The study was also conducted in accordance with the provisions of the Declaration of Helsinki and the Ethical Guidelines for Epidemiological Studies issued by MHLW, Japan. As per the ethical guidelines, informed consent is not required if it is necessary for public health and uses anonymized data. Personal information was removed from all data by a unique anonymization process performed by the community integrated care section of Ikoma city, which kept researchers blinded to all participant personal data.

### Data collection

Data were extracted from the KCL and the long-term care database managed by the community integrated care section of the city of Ikoma.

### Assessment of functional decline in daily living and frailty assessment

For the assessment of functional decline in daily living, we used the KCL, which included not only physical activities but also psychological, mental, and instrumental activities of daily living (IADL). The KCL is composed of 25 questions in 7 areas and answers can indicate functional decline (S1 Fig). Assessment of functional decline in each area is determined by the following: ≥3 of 5 motor function items (question number 6 to 10), both nutritional status items (question number 11 and 12), ≥2 of 3 oral function items (question number 13–15), homebound status (question number 16), ≥1 of 3 cognitive function items (question number 18–20), and ≥2 of 5 depressive mood items (question number 21–25). In this study, IADL decline was defined as any of the 5 social activity items that corresponded to any of 3 IADL items (question number 1 to 3). Question number 4 and 17 are used for the overall score of the frailty identification, but these are not used for the judgment of "homebound status" and "decrease in IADL" in the KCL assessment procedure. The frailty assessment criteria are based

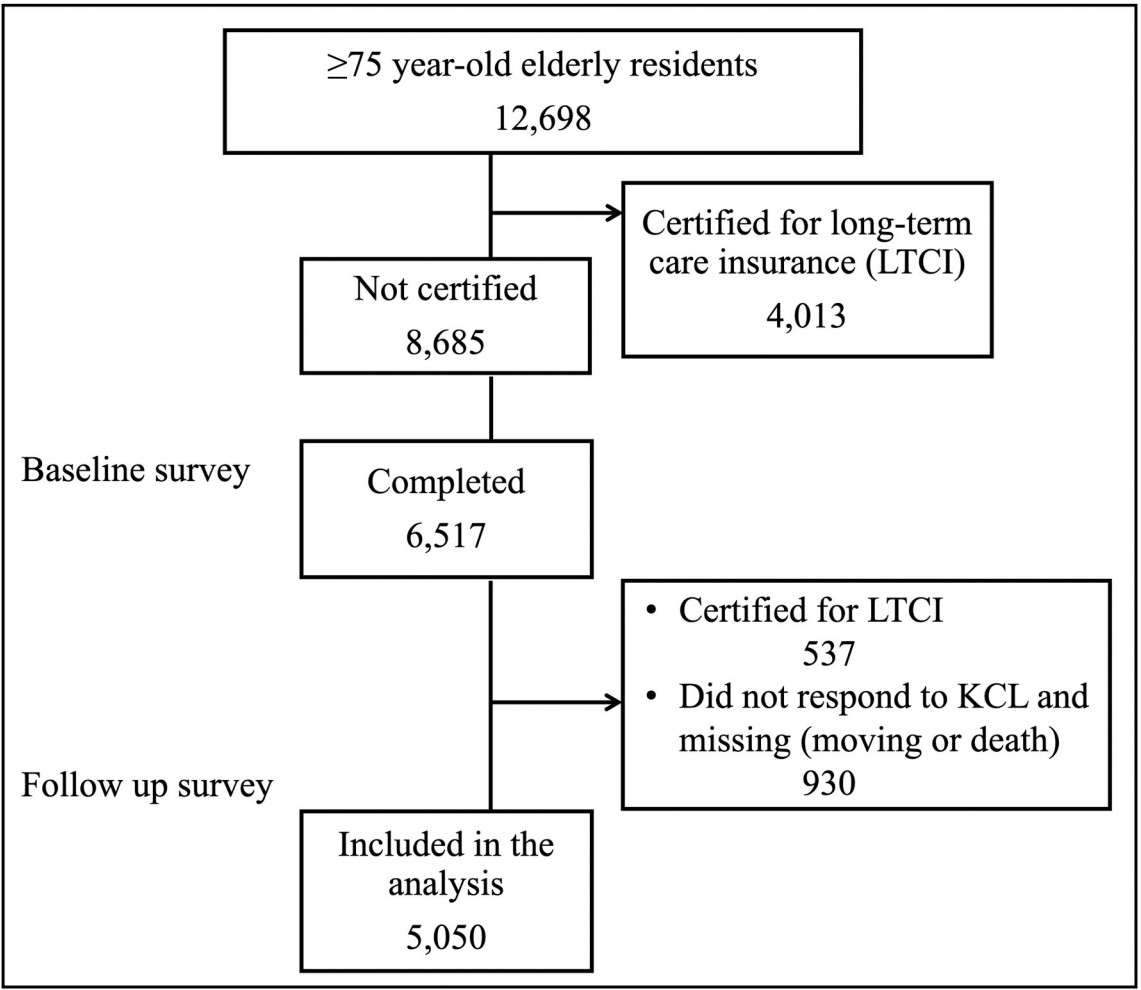

**Fig 1. Flow chart of participant selection.**

on the number of checked items in the 25 questions consisting of 7 areas, according to previous research [22]: 0–3 for robust, 4–7 for pre-frailty, and ≥8 for frailty.

Regarding recovery from frailty, the change in the frailty assessment between the baseline survey and the follow-up survey 2 years later was compared, and the number of people who improved from frailty to robust or pre-frailty was determined (not including change from pre-frailty to robust).

## Assessment of other variables

In addition to KCL, other assessments included items related to individual-level social capital strength, such as social participation activities, the degree of trust in the community, and the degree of interaction with neighbors.

Community activities include participation in senior citizen clubs, residents' associations, and elderly salons. Exercise-based activities refer to regular participation in gymnastics classes, ground golf (Japanese style golf suitable for seniors), Japanese croquet, and other activities engaged in by local residents. Hobby activities include non-exercise activities such as handicrafts, gardening, and board games, while volunteer/NPO activities include community

cleaning activities and counselling of neighboring residents. On the questionnaire, the presence or absence of the above-mentioned social participation activities and the total number of activity categories were recorded as social participation scores.

The level of trust in the community was evaluated on a scale of 5 from "very important" to "not important" in response to the question "How much do you think trust in your neighborhood is important to everyday life?". The degree of interaction with neighbors was rated on a 4-point scale from "There are people who talk with each other and cooperate in terms of life" to "no interaction with neighbors". Self-rated health was assessed on a 5-point scale according to a previous study [23]. Subjective age was evaluated in the form of "yes" or "no" in response to the question "Do you feel younger than your actual age?" With regard to subjective age, previous studies have shown that being subjectively younger than one's chronological age (actual age) has a positive effect on various health-related outcomes such as physical function, cognitive functions, depressive symptoms, and life satisfaction of the elderly [24–27]. Therefore, in this study, it was included in the evaluation item as a factor that has the potential to prevent frailty progression.

## Covariates

Sociodemographic characteristics (age, sex, living alone) and the number of chronic diseases (hypertension, sequelae of stroke, heart disease, diabetes, depression, respiratory disease, arthropathy with pain, and dental disease) were assessed as covariates. The number of chronic diseases in each participant was defined as the disease burden.

## Statistical analysis

First, for group comparison based on frailty assessment at the baseline survey, $\chi^2$-test with residual analysis was used for categorical variables, and one-way analysis of variance (multiple comparison test: LSD test) was used for continuous variables. Next, the distribution of frailty over 2 years was cross-tabulated, and changes in states were analyzed by $\chi^2$-test and adjusted residual test. We considered having significantly more participants than expected when the adjusted residual values were higher than 1.96, while having significantly fewer participants than expected when the adjusted residual values were lower than -1.96. Third, in order to extract factors that reverse frailty progression at 2 years after the baseline survey, multiple logistic regression analysis was performed with the presence or absence of recovery from frailty as a dependent variable. Recovery from frailty in this study was defined as a change of state from frailty to pre-frailty or robust. For the independent variables, self-rated health, subjective age (young or old), the presence or absence of each social participation activities (community-based activity, exercise-based activity, hobby activity, volunteer or NPO activity), strength of interaction with neighbors, and strength of trust in neighbors were imputed using the forced entry method. In addition, variables were adjusted by sex, age, living alone or not, and disease burden. Finally, the participants who recovered from frailty in the follow-up survey were extracted as 1 dataset. After that, principal component analysis to clarify the characteristics of the participants who recovered from frailty was performed by inputting all variables except those with binary results (i.e., subjective age, history of fall, and fear of falling). For social participation activities, the total number of participating events was used as a score. Principal components were obtained up to components having Eigenvalues $\geq$1.0. Analyses were carried out using SPSS software (version 26.0; SPSS, Chicago, IL, USA).

## Results

A total of 5050 participants were included for statistical analysis. Table 1 summarizes the demographic characteristics of all participants as well as participant characteristics subgrouped by frailty states. At baseline, the mean ± SD age of all participants was 79.4 ± 3.8 years

**Table 1. Participant characteristics and frailty classification at baseline.**

| Items | All | Robust | Pre-frailty | Frailty | P | Between subgroup difference |
|---|---|---|---|---|---|---|
| | (n = 5050) | (n = 2306) | (n = 1802) | (n = 942) | | |
| Age, y (SD) | 79.4 (3.8) | 78.6 (3.3) | 79.7 (3.2) | 80.8 (4.4) | <0.001 | Frail>Pre-frail>Robust |
| Sex: female, n (%) | 2538 (50.3) | 1002 (43.5) | 978 (54.3) | 558(59.2) | <0.001 | Frail>Pre-frail>Robust |
| Disease burden (SD) | 1.4 (0.9) | 1.2 (0.9) | 1.4 (0.9) | 1.7 (1.0) | <0.001 | Frail>Pre-frail>Robust |
| Living alone, n (%) | 765 (19.1) | 330 (14.3) | 269 (14.9) | 166 (17.6) | 0.053 | n.s |
| IADL decline, n (%)[†] | 210 (4.2) | 2 (0.1) | 7 (0.4) | 201 (21.3) | <0.001 | Frail>Pre-frail>Robust |
| Motor dysfunction, n (%)[†] | 812 (16.1) | 17 (0.7) | 292 (16.2) | 503 (53.4) | <0.001 | Frail>Pre-frail>Robust |
| Malnutrition, n (%)[†] | 103 (2.0) | 15 (0.7) | 32 (1.8) | 56 (5.9) | <0.012 | Frail>Robust |
| Oral dysfunction, n (%)[†] | 885 (17.5) | 50 (2.2) | 359 (19.9) | 476(50.5) | <0.001 | Frail>Pre-frail>Robust |
| Homebound, n (%)[†] | 274 (5.4) | 25 (1.1) | 74 (4.1) | 175 (18.6) | <0.001 | Frail>Pre-frail>Robust |
| Cognitive decline, n (%)[†] | 1515 (30.0) | 300 (13.0) | 633 (35.1) | 582 (61.8) | <0.001 | Frail>Pre-frail>Robust |
| Depressive mood, n (%)[†] | 1357 (26.9) | 44 (1.9) | 389 (21.6) | 706 (74.9) | <0.001 | Frail>Robust |
| History of fall, n (%)[†] | 974 (19.3%) | 140 (6.1) | 360 (20.0) | 351 (37.4) | <0.001 | Frail>Pre-frail>Robust |
| Fear of falling, n (%)[†] | 2463 (48.8) | 391 (17.3) | 948 (52.6) | 752 (80.8) | <0.001 | Frail>Pre-frail>Robust |
| Self-rated health (SD) | 3.5 (1.7) | 3.9(1.0) | 3.3(0.9) | 2.7 (0.9) | <0.001 | Robust>Pre-frail>Frail |
| Subjective age (young), n (%) | 4001 (87.0) | 2041 (94.8) | 1424 (87.2) | 536(65.8) | <0.001 | Robust>Pre-frail>Frail |
| Social participation score (SD) | 0.9 (1.0) | 1.0 (1.1) | 0.8 (0.9) | 0.5 (0.7) | <0.001 | Robust>Pre-frail>Frail |
| Interaction with neighbors (SD) | 3.0 (1.8) | 2.0 (0.7) | 1.8 (0.7) | 1.6 (0.8) | <0.001 | Robust>Pre-frail>Frail |
| Trust in neighbors (SD) | 4.0 (1.1) | 4.1 (1.0) | 4.0 (1.2) | 3.8 (1.2) | <0.001 | Robust>Pre-frail>Frail |

†Based on Kihon Checklist sub-score of each area. ‡Frailty identification: Out of KCL 25 items, 0–3 for robust, 4–7 for pre-frailty, and >8 for frailty. SD, standard deviation. Categorical variables were analyzed by χ2-test (with residual test), and continuous variables were analyzed by one-way ANOVA (with post-hoc LSD test).

(range, 75–99 years), and 50.3% were female. The mean disease burden was $1.4 \pm 0.9$ (range, 0–6). According to the frailty assessment, 2306 (45.7%) were robust, 1802 (35.7%) had pre-frailty, and 942 (18.7%) had frailty. Comparing the subgroups, the frailty group was older, had a higher proportion of females, and had a higher disease burden than the other groups (each $P<0.001$). In addition, except for the ratio of living alone, the frailty group had significantly lower life function, motor function, mental/psychological status, and social participation activity than the other groups (each $P<0.001$).

Fig 2 shows the changes in frailty states after 2 years. The recovery rate from frailty at 2 years was 31.8% (n = 300), which accounted for 5.9% of all participants. As indicated by $\chi^2$-test with residual analysis, the change in frailty state was mostly maintained regardless of the frailty classification at baseline, and the change to deterioration or improvement was small (Table 2). Results of multiple logistic regression analysis adjusted for age, sex, living alone, and disease burden showed that exercise-based social participation (odds ratio [OR] 2.0, 95% confidence interval [CI] 1.2–3.4; $P<0.01$) and good self-rated health (OR 1.2, CI 1.0–1.5; $P = 0.02$) were related to reversing frailty progression (Table 3). In an additional analysis using the same adjustment variables, when the independent variable was input with or without both factors (good self-rated health and exercise-based social participation), those who had both factors were more likely, than those who did not, to recover from frailty (OR 1.9, 95%CI 1.5–2.6; $P<0.01$).

Table 4 shows demographic characteristics, results of the KCL assessment, and other items of those who recovered from frailty. Despite some participants showing reversal of frailty progression, the proportion of those with cognitive decline and fear of falling was higher when compared with the baseline data.

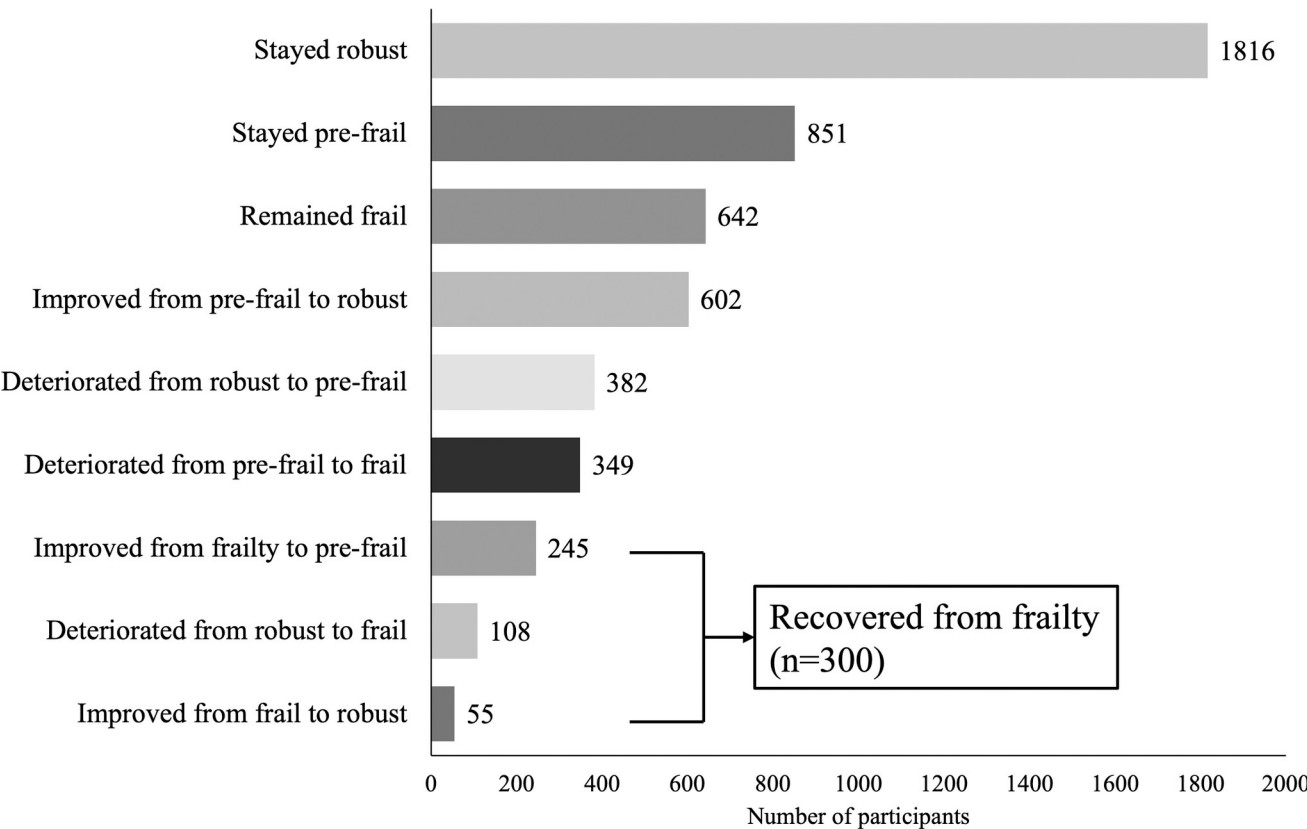

**Fig 2. Changes in frailty states after 2 years.**

Principal component analysis, performed to clarify the factor structure of participants who recovered from frailty, revealed the 4 principal components, and the cumulative contribution

**Table 2. Cross tabulation of frailty states after 2 years.**

| | | | Follow-up frailty assessment | | | Total |
|---|---|---|---|---|---|---|
| | | | Robust | Pre-frailty | Frailty | |
| Baseline frailty assessment | Robust | Observed | 1816 | 382 | 108 | 2306 |
| | | Adjusted residual | 38.8 | -18.2 | -27.0 | |
| | Pre-frailty | Observed | 602 | 851 | 349 | 1802 |
| | | Adjusted residual | -16.5 | 20.9 | -3.1 | |
| | Frailty | Observed | 55 | 245 | 642 | 942 |
| | | Adjusted residual | -29.4 | -2.4 | 38.3 | |
| Total | | | 2473 | 1478 | 1099 | 5050 |
| Chi-square test | | | | | | |
| | | | Value | df | P | |
| Pearson chi-square | | | 2439.091[a] | 4 | 0.000 | |
| Likelihood ratio | | | 2405.336 | 4 | 0.000 | |
| Linear-by-linear association | | | 2029.236 | 1 | 0.000 | |
| Number of valid cases | | | 5050 | | | |

df, degrees of freedom

**Table 3. Factors associated with reversing frailty progression (n = 942).**

| Items | P | OR | 95%CI |
|---|---|---|---|
| Self-rated health | 0.02 | 1.24 | 1.04–1.49 |
| Subjective age (young) | 0.23 | 1.25 | 0.87–1.79 |
| Social participation activity | – | – | – |
| • Community-based activity | 0.55 | 1.15 | 0.73–1.79 |
| • Exercise-based activity | 0.00 | 2.02 | 1.19–3.42 |
| • Hobby activities (cultural) | 0.83 | 0.95 | 0.61–1.48 |
| • Volunteer/NPO activity | 0.14 | 1.77 | 0.81–3.87 |
| Interaction with neighbors | 0.24 | 1.14 | 0.92–1.42 |
| Trust in neighbors | 0.89 | 1.01 | 0.88–1.17 |

Adjusted for sex, age, living alone or not, disease burden. † Independent variables were input from the results of baseline data. OR, odds ratio; CI, confidence interval.

rate was 54.1% (Table 5). The KCL sub-score was inverted and input for the analysis so that the smaller the functional decline, the higher the score. The main variables that constitute the first principal component of the commonalities of the observed information were interaction with neighbors, trust in neighbors, social participation score, and IADL (contribution rate 18.3%). From these variables, the construct of the first principal component was interpreted as "the strength of individual-level social capital for living active community life."

## Discussion

Although the concept of frailty is based on reversibility, many previous studies have focused on risk factors and protective factors for frailty [28]. Many studies have identified risk factors for frailty, including age, sex (female), cognitive ability (executive function, etc.), and

**Table 4. Participants who recovered from frailty (n = 300).**

| Items | Mean (SD) or number (%) |
|---|---|
| Age, y | 79.9 (4.0) |
| Sex: female, n (%) | 163 (54.3) |
| Disease burden (SD) | 1.6 (1.0) |
| IADL decline, n (%)[†] | 57 (19) |
| Motor dysfunction, n (%)[†] | 58 (19.8) |
| Malnutrition, n (%)[†] | 7 (2.3) |
| Oral dysfunction[†] | 71 (23.7) |
| Homebound, n (%)[†] | 23 (7.7) |
| Cognitive decline, n (%)[†] | 131 (43.7) |
| Depressive mood, n (%)[†] | 85 (28.3) |
| History of fall, n (%) [†] | 49 (16.3) |
| Fear of falling, n (%)[†] | 193 (64.3) |
| Self-rated health (SD) | 3.2 (0.9) |
| Subjective age (young), n (%) | 264 (88.0) |
| Social participation score (SD) | 0.7 (0.9) |
| Interaction with neighbors (SD) | 1.7 (0.7) |
| Trust in neighbors (SD) | 3.9 (0.9) |

[†]Results in each area of the Kihon checklist were based on the follow-up survey data. SD, standard deviation.

**Table 5. Principal Component Analysis (PCA) results to elucidate characteristics of participants who recovered from frailty.**

| Variable | PCA factor | | | |
|---|---|---|---|---|
| | 1 | 2 | 3 | 4 |
| Interaction with neighbors | **0.689** | 0.322 | -0.192 | 0.066 |
| Trust in neighbors | **0.669** | 0.186 | 0.168 | 0.253 |
| Social participation activity | **0.666** | -0.071 | -0.082 | 0.333 |
| IADL[†] | **0.503** | -0.41 | -0.056 | -0.421 |
| Depressive mood[†] | 0.002 | **0.689** | -0.305 | 0.086 |
| Nutritional state[†] | 0.003 | 0.103 | **0.631** | 0.272 |
| Homebound[†] | 0.275 | -0.435 | **0.445** | 0.073 |
| Oral function[†] | -0.263 | 0.289 | **0.427** | 0.325 |
| Motor function[†] | -0.057 | -0.37 | -0.357 | **0.578** |
| Cognitive function[†] | 0.258 | 0.308 | 0.276 | **-0.496** |
| % variance explained | 18.35 | 13.02 | 11.5 | 11.29 |

Values are calculated from the results of follow-up surveys.

[†]Each variable was calculated by inverting the value so that the smaller the functional decline, the higher the score.

socioeconomic factors (financial stress, etc.) [28]. Conversely, the rate of recovery from frailty has not been investigated sufficiently, and the manifestation of frailty has been limited to physical frailty [29–31]. The reversibility of frailty applies not only to physical frailty but also to mental and psychological frailty including mild cognitive impairment (MCI) and social frailty. Most factors that contribute to physical frailty can be improved by appropriate intervention, and social frailty is also comprised of factors that can be changed through behavioral change, except for factors such as living alone.

Regarding the importance of focusing on social frailty, Tanaka et al. [32] compared people with and without social frailty in a 5-year cohort study that excluded those who were physically frail at baseline, homebound, living alone, or lacking family support. The incidence of and mortality rate from disability was shown to be about 3 times higher in those with versus without social frailty in a baseline survey. In addition, Tsutsumimoto et al. [33] reported that the proportion of people with social frailty increased among those aged 75 years and over, and social frailty was independently associated with decreased physical function (grip strength, walking speed) and cognitive function (memory, calculation, attention). Accordingly, several measurement methods have been developed for the evaluation method of social frailty, and, in addition to the above-mentioned factors, the relationship with QOL and IADL independence has been reported [34, 35]. Therefore, the assessment of frailty should be a comprehensive assessment that includes mental and psychological aspects, social aspects, and cognitive function, and should be applied especially to late-stage elderly people with a high risk of requiring long-term care. Many previous studies have developed various assessment tools for frailty, but the KCL is a comprehensive assessment that includes the above mentioned aspects. Frailty judgment by KCL has also shown predictive validity for the future need of nursing care [14]. In Japan, all municipalities have announced that they will begin implementing frailty medical checkups for people ≥75 years old beginning in 2020 [36].

The results of this study showed that the recovery rate from frailty was 31.8%. Our definition of reversing frailty progression included the change from frailty to pre-frailty and, when the focus was on the 2-stage changes from frailty to robust, which is considered to be the true recovery rate, the rate was only 5.8%. Gill and colleagues surveyed longitudinally the transition from frailty in 754 community-dwelling people aged ≥70 years every 18 months for 4.5 years

and reported that the probability of transition from frailty to robust was 0%-0.9% [31]. Although the participants' age and observation periods were different, the transition rates in our study were superior to their results. On the other hand, the results of a survey of 420 women aged 70–79 years from the Women's Health and Aging Studies (WHAS) II showed that the rate of transition from frailty to non-frailty was noticeably higher (17%) during the first 18 months compared with the rate in a previous study [29]. However, their study is difficult to compare with our findings because their study participants consisted of females only, the participants had a different age range, the study focused only on physical frailty, and the sample size was smaller than that in our study. As described above, the recovery rate from frailty varies among studies, but the common point between previous research and our findings is that a 2-stage improvement from frailty to robust is very rare.

Focusing on the improvement from pre-frailty to robust, the improvement rate in this study was 33.4%, which was more than 6 times the rate of transition from frailty to robust. Gill and colleagues reported a 16.9%-35.0% improvement from pre-frailty to robust over an 18-month observation period [30]. In their report, the improvement rate fluctuated depending on the observation period, but in terms reversibility, their data suggest the importance of early detection of pre-frailty and intervention, as did the results of our study.

In many cases, frailty research focuses on risk factors, as physical frailty often develops in a short period of time without going through a pre-frailty stage [29, 30]. For this reason, little attention has been paid to factors that assist in reversing frailty progression. In a cross-sectional study conducted in Japan, walking more than 30 minutes a day, meeting friends more than once a month, and going out every day were cited as factors for reversing frailty progression [21]. The results of our study indicate that high self-rated health and participation in exercise-based social activity (i.e., gymnastics classes, Japanese croquet, etc.) are independent factors that influence reversing frailty progression. These results suggest that the interaction between the practice of habitual exercise and the improvement of subjective health is important for reversing frailty progression. In general, since late-stage elderly people have some chronic disease, the magnitude of the disease burden was included in the logistic model, but no independent effect was observed on recovery from frailty. The reason for this is thought to be that those with diseases that affect their daily lives shifted to long-term care insurance services during the follow-up period.

Principal component analysis using a data subset from those who recovered from frailty showed that interaction with neighbors, trust in neighbors, social participation activities, and IADL scores were the main factors that constituted the first principal component. Given the commonality of each factor, we named the construct of the first principal component "the strength of individual-level social capital for living active community life." Based on the results of the logistic regression analysis and the principal component analysis, elderly people who reversed frailty progression have strong relations with the neighborhood, which may have improved their physical functions and psychological aspects. In recent years, there has been growing interest in social frailty for frailty classification, and it has been reported that social frailty assessment based on items such as living alone and not talking to someone every day predicts the need for nursing care [8, 33]. The results of our study indicate that the premise of physical and psychological improvement in older adults is the need to connect with their neighborhoods in everyday life.

In Japan, by 2025, when baby boomers will be ≥75 years old, a structure called the "Community-based Integrated Care System" will be established that comprehensively ensures the provision of health care, nursing care, prevention, housing, and livelihood support [37]. In preventing the need for long-term care, the promotion of "self-help and mutual help" among the elderly is emphasized. Therefore, it is important to increase the number of small-scale

exercise classes sponsored by residents within walking distance of local elderly people in each small district, and to create a system in which healthy early-stage elderly people support late-stage elderly people. Furthermore, it is indispensable to involve professionals in various fields such as exercise, nutrition, and long-term care to regularly check changes in physical condition and provide appropriate educational guidance.

The present study has some limitations. First, since this was an observational study, it is not clear which factors affected behavioral changes. Second, the participants who returned the KCL were a group with good health awareness and good cognitive function. Finally, in the principal component analysis, the cumulative contribution ratio was small, and there are many factors that cannot be explained by the assessment items in this study.

In the future, we will continue the follow-up survey and analyze the rate of recovery from frailty based on social capital intensity among small regions, and the relationship with the presence and type of social support (instrumental support, emotional support).

## Conclusions

Participation in exercise-based social activities is independently associated with reversing frailty progression, and people who recovered from frailty are characterized by high social capital components, such as trust in the community and interaction with their neighbors. Thus, to increase the reversibility from frailty, residents may be advised to engage in initiatives in their district to increase their opportunities to participate in exercise-based social activity for the elderly and to enhance mutual help by interacting with each other.

## Supporting information

**S1 Fig. English version of the Kihon checklist.**
(TIF)

## Acknowledgments

We thank the community integrated care section of Ikoma city for their cooperation with participant recruitment and data provision.

## Author Contributions

**Conceptualization:** Katsuhiko Takatori, Daisuke Matsumoto.

**Data curation:** Katsuhiko Takatori.

**Formal analysis:** Katsuhiko Takatori.

**Investigation:** Katsuhiko Takatori, Daisuke Matsumoto.

**Methodology:** Katsuhiko Takatori, Daisuke Matsumoto.

**Project administration:** Katsuhiko Takatori.

**Resources:** Katsuhiko Takatori, Daisuke Matsumoto.

**Supervision:** Katsuhiko Takatori.

**Validation:** Daisuke Matsumoto.

**Writing – original draft:** Katsuhiko Takatori.

**Writing – review & editing:** Daisuke Matsumoto.

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
