## [Decision Letter · Decision Letter 0]

13 Oct 2020

PONE-D-20-26897

Factors that assist escape from frailty in community-dwelling late-stage elderly people: An observational study

PLOS ONE

Dear Dr. Takatori,

Thank you for submitting your manuscript to PLOS ONE. After careful consideration, we feel that it has merit but does not fully meet PLOS ONE’s publication criteria as it currently stands. Therefore, we invite you to submit a revised version of the manuscript that addresses the points raised during the review process.

We look forward to receiving your revised manuscript.

Kind regards,

Simone Reppermund, PhD

Academic Editor

PLOS ONE

Journal Requirements:

"NO"

Reviewers' comments:

Reviewer's Responses to Questions

**Comments to the Author**

1. Is the manuscript technically sound, and do the data support the conclusions?

Reviewer #1: Partly

Reviewer #2: Yes

2. Has the statistical analysis been performed appropriately and rigorously? 

Reviewer #1: Yes

Reviewer #2: Yes

3. Have the authors made all data underlying the findings in their manuscript fully available?

Reviewer #1: Yes

Reviewer #2: Yes

4. Is the manuscript presented in an intelligible fashion and written in standard English?

Reviewer #1: No

Reviewer #2: Yes

5. Review Comments to the Author

Reviewer #1: Factors that assist escape from frailty in community-dwelling late-stage elderly people:

An observational study

Takatori et al. describe an observational cohort study over a 2-year period examining healthy lifestyle habits and participant characteristics that are associated with reversing frailty progression in Ikoma city, Japan. The authors should be commended for coordinating data collection on a large number of participants on a topic area that is becoming increasingly important – finding methods of preventing or reversing frailty progression. Unfortunately, the authors do not present a sound understanding of frailty in their introduction nor a sound literature review on frailty intervention to date, no hypotheses were presented and the methodology description in its current state would make repeating this study very difficult. The manuscript writing could also use revaluation as it was fraught with word usage that I would not recommend throughout, such as “escape from frailty” and "senile decay". Unclear if the STROBE checklist was used in the creation of this manuscript.

Introduction

- the presented definition of frailty is not one that is commonly used in the field and seminal pieces of work on frailty (such as those by Fried and Rockwood) are not used in the definition discussion. What is described is far from what would be considered the current characterization of frailty and the line of reasoning presented to explain their description is hard to follow.

- "escape from frailty" is not terminology I would use. Instead, you are discussing reversing frailty progression. This becomes much easier to comprehend and more intuitively also describes a transition from frailty to pre-frailty.

- subjective age was presented in the introduction but not described until much later. A better descriptor of what the authors were actually discussing is a 'younger subjective age' or similar that would better represent what is being discussed.

- As a less used measure of frailty, providing more data on KCL correlation to more widely accepted methods would be recommended.

- “senile decay”??? (line 89) – better description of why reversing or slowing frailty progression is important

- There have been a number of intervention studies that could have been discussed in the background literature including best recommendations thus far. Only one was mentioned – this was not an extensive literature review by any means.

- Clearer description needed of “factors that assist” in reversing frailty and distinguishing those from “characteristics" of those who see frailty reversibility.

- No hypothesis is presented.

Methods

- How was the study population found?

- Flow chart of participant progression was unclear. “Certified”? – how was ‘certification’ determined. Flow chart talks about insurance? Why would participants be dropped if they now required LTCI through the follow up period? I thought this was the authors' definition of frailty, albeit one that I disagree with.

- Frailty 'escapers' is not well described. Instead what is being described is reversing frailty progression.

- Examining those who did not have their frailty progress would also have been interesting.

Results

- Describe sex but use term “women” – women refers to gender, female refers to sex

- Is ‘escape from frailty’ just those who went from frail to robust or also those who went from pre-frail to robust? This seems to only be described in the discussion.

- I would recommend presenting change in frailty status differently graphically and statistically. With a checklist with many variables like the KCL, this data could have been presented in a continuous fashion as opposed to categorical changes which result in a loss of information.

Discussion

- Definition of 'escape from frailty' only provided in the discussion?

- Authors start on good discussion of relative reversibility of frail vs pre-frail states but don’t follow through on description of why this might be or references to seminal works.

- Authors should consider if ‘social frailty’ is really a distinct construct from frailty if these factors are included in their frailty assessment. Is this not just a subdomain of frailty then? Are external factors like these social ones discussed actually measures of frailty, or are they associated factors that could precipitate frailty development?

- What do we do with this information? Authors do not provide recommendations for how to incorporate findings into any kind of practice.

- Conclusions have to be more strongly discussed as being factors associated with reversing frailty as this assessment is not an intervention.

Reviewer #2: Major comments:

The authors collect disease burden in the present study, which appear to not be a part of their frailty measure. How might an individual’s disease burden affect their “escape” from frailty?

Here the authors only assess “escape from frailty” However it would be useful to determine if there is worsening frailty among the group, and which factors might lead to that.

It would be interesting to determine the cumulative effect of factors examined here on escape from and transition to frailty (and those who stay stable). For example, do the number of factors an individual has have a greater odds of transitioning to a better state?

Minor comments:

Please include in the abstract the follow up time period (median preferable)

Please include in the abstract more detail about the Kihon checklist (i.e., how many items, range scores, what indicates frailty)

Title is misleading regarding “factors” that assist escape from frailty. It appears that the authors focus on specific factors which are predominantly social factors – this needs to be made clear

I think it would be useful to re review the manuscript for English language grammar/spelling/comprehension.

Please provide additional information on how participants were selected for receiving a survey. Was this a randomized process? Convenience sampling?

6. PLOS authors have the option to publish the peer review history of their article (what does this mean?). If published, this will include your full peer review and any attached files.

Reviewer #1: No

Reviewer #2: **Yes: **Dustin Scott Kehler

---

## [Author Response · Author response to Decision Letter 0]

24 Nov 2020

The authors received no specific funding for this work.

---

## [Decision Letter · Decision Letter 1]

7 Jan 2021

PONE-D-20-26897R1

Social factors that reverse frailty progression in community-dwelling late-stage elderly people: An observational study

PLOS ONE

Dear Dr. Takatori,

Thank you for submitting your manuscript to PLOS ONE. After careful consideration, we feel that it has merit but does not fully meet PLOS ONE’s publication criteria as it currently stands. Therefore, we invite you to submit a revised version of the manuscript that addresses the points raised during the review process.

We look forward to receiving your revised manuscript.

Kind regards,

Simone Reppermund, PhD

Academic Editor

PLOS ONE

Reviewers' comments:

Reviewer's Responses to Questions

**Comments to the Author**

1. If the authors have adequately addressed your comments raised in a previous round of review and you feel that this manuscript is now acceptable for publication, you may indicate that here to bypass the “Comments to the Author” section, enter your conflict of interest statement in the “Confidential to Editor” section, and submit your "Accept" recommendation.

Reviewer #1: All comments have been addressed

Reviewer #2: All comments have been addressed

2. Is the manuscript technically sound, and do the data support the conclusions?

Reviewer #1: Yes

Reviewer #2: Yes

3. Has the statistical analysis been performed appropriately and rigorously? 

Reviewer #1: Yes

Reviewer #2: Yes

4. Have the authors made all data underlying the findings in their manuscript fully available?

Reviewer #1: Yes

Reviewer #2: Yes

5. Is the manuscript presented in an intelligible fashion and written in standard English?

Reviewer #1: No

Reviewer #2: Yes

6. Review Comments to the Author

Reviewer #1: Takatori et al. describe an observational cohort study over a 2-year period examining healthy lifestyle habits and participant characteristics that are associated with reversing frailty progression in Ikoma city, Japan. Thank you to the authors for addressing the suggested revisions in their manuscript. The authors addressed the suggested revisions, but the manuscript still requires work to improve clarity. The manuscript also requires thorough review of the English throughout for grammar and clarity.

-The title is much more descriptive of the article now but the way it is currently written implies a causative finding despite the identification of it being an observational study. Please edit the title so that it implies associations as the findings.

-It would be beneficial to review the English throughout the publication for grammar and clarity.

Abstract

-Line 29: Are those in nursing care not frail? The authors’ current first sentence would imply this, but this is not true. Based on the description from the Japanese Geriatric Society, frailty status can be associated with adverse outcomes like resorting to nursing care, but the description of frailty as being a state of health vulnerability as described late in the introduction is much more appropriate for use in the opening of the abstract than what is currently written.

-Line 42: It would be easier for the reader to interpret participant prevalence here by listing the recovery rate as a percentage of those who actually had frailty. Those who did not have frailty at baseline could not possible have reversed their frailty status.

-Line 49: Results presented should not be discussed as causative finding. This was an observational study. These should be identified as associations.

Introduction

-Again, the statement from the Japanese Geriatric Society is not being interpreted correctly in lines 60 and 61. They say it is vulnerability that leads to adverse outcomes such as need for nursing care. It would be incorrect to then imply that those receiving nursing care are not frail. That is just one of the adverse outcomes associated with frailty due to the health vulnerability it is characterized by.

-Line 64: Please provide a citation to support this change in demographic trend.

-Line 71: The end of this sentence should say “…were pre-frail.”

-Line 76: This is an interesting citation to select. There are a number of more recent consensus statements regarding frailty.

-Line 79: I do not think it is apt to say that one would diagnose frailty. Frailty is not a disease and not a commonly identified outcome in clinical practice, thus it would be more appropriate to say ”…for frailty identification.”

-Line 82-83: Accumulation of Deficits Models

-Line 85: “Frailty assessment” instead of “judgement”

-Lines 94-97: It is unclear what the authors are trying to state and the English here could use a thorough review.

-Line 93-94: The English in these two sentences needs to be reviewed. It is difficult to understand what this sentence is trying to present. Additionally, no citations are provided to support the statement made.

-Line 98-99: Citation required for this statement.

-Line 100-101: Please provide citations associated with the statements made here.

-Line 101-102: What reports have identified these factors? What factors were identified? Please provide citations here as well.

-Line 108: “…in improving frailty status”

-Lines 108-111: This sentence needs to be rewritten for clarity.

Methods

-Line 142; …frailty assessment

Results

-Fig 2: The English use here could use some review. “stayed robust”, “stayed pre-frail”, “remained frail”

Discussion

-Line 284: No citations provided to support this statement.

-Line 287: No citations provided to support this statement.

-Line 295: “No-frailty people”?

-Line 312-317: A lot of this text was already included in the results section. Please make this more succinct and focused on discussion aspects.

-Line 337-338: Unclear what this sentence is stating.

-Line 342: Please provide citations to support this statement.

Conclusions

-Again, these findings are stated quite strongly for an observational study. It should be clear that these findings are associations.

Reviewer #2: The authors have adequately addressed the reviewer comments. I appreciate the additional analyses that were performed.

7. PLOS authors have the option to publish the peer review history of their article (what does this mean?). If published, this will include your full peer review and any attached files.

Reviewer #1: No

Reviewer #2: No

---

## [Author Response · Author response to Decision Letter 1]

2 Feb 2021

Authors' Response to Reviewers

For Reviewer #1

Thank you for your valuable comments and your appreciation of the content of our manuscript. We have replied to your comments point-by-point below.

This time, we asked a proofreader to review the English expression and grammar in general, and revised it to improve the clarity of the content. 

Title 

- The title is much more descriptive of the article now but the way it is currently written implies a causative finding despite the identification of it being an observational study. Please edit the title so that it implies associations as the findings.

We changed the wording that suggests a causal relationship and revised the title.

Abstract

- Line 29: Are those in nursing care not frail?

- The authors’ current first sentence would imply this, but this is not true. Based on the description from the Japanese Geriatric Society, frailty status can be associated with adverse outcomes like resorting to nursing care, but the description of frailty as being a state of health vulnerability as described late in the introduction is much more appropriate for use in the opening of the abstract than what is currently written.

We have modified the wording about the definition of frailty at the beginning.

- Line 42: It would be easier for the reader to interpret participant prevalence here by listing the recovery rate as a percentage of those who actually had frailty. 

- Those who did not have frailty at baseline could not possible have reversed their frailty status. 

We have re-calculated the recovery rate of the number of participants that have been classified as frailty at baseline as the denominator.

- Line 49: Results presented should not be discussed as causative finding. This was an observational study. These should be identified as associations.

We have changed the wording about causality (Line 51).

Introduction

- Again, the statement from the Japanese Geriatric Society is not being interpreted correctly in lines 60 and 61.

- They say it is vulnerability that leads to adverse outcomes such as need for nursing care. It would be incorrect to then imply that those receiving nursing care are not frail. 

- That is just one of the adverse outcomes associated with frailty due to the health vulnerability it is characterized by.

In addition to modifying the abstract, we have removed the misleading part of the definition of frailty.

- Line 64: Please provide a citation to support this change in demographic trend.

We have added one new citation (Ref No.5).

- Line 71: The end of this sentence should say “…were pre-frail.”

We modified it according to the comment (Line 72).

- Line 76: This is an interesting citation to select. There are a number of more recent consensus statements regarding frailty.

We have changed to a newer citation (Ref No.7).

- Line 79: I do not think it is apt to say that one would diagnose frailty. Frailty is not a disease and not a commonly identified outcome in clinical practice, thus it would be more appropriate to say ”…for frailty identification.”

We modified it according to the comment (Line 80).

- Line 82-83: Accumulation of Deficits Models

We modified it according to the comment (Line 84).

- Line 85: “Frailty assessment” instead of “judgement”

We modified it according to the comment (Line 86).

- Lines 94-97: It is unclear what the authors are trying to state and the English here could use a thorough review.

- Line 93-94: The English in these two sentences needs to be reviewed. It is difficult to understand what this sentence is trying to present. Additionally, no citations are provided to support the statement made.

We decided to delete this part because we thought that this part was not important in describing the background of our research.

- Line 98-99: Citation required for this statement.

- Line 100-101: Please provide citations associated with the statements made here.

We have added one new citation (Ref No.19).

- Line 101-102: What reports have identified these factors? What factors were identified? Please provide citations here as well.

We have added a new citation and added factors that have been identified as improving factors from frailty (Ref No.20, Line 100-102).

- Line 108: “…in improving frailty status”

We modified it according to the comment (Line 110).

- Lines 108-111: This sentence needs to be rewritten for clarity.

We deleted some sentences and modified the wording (Line 110-112).

Methods

- Line 142; …frailty assessment

We modified it according to the comment (Line 142).

Results

- Fig 2: The English use here could use some review. “stayed robust”, “stayed pre-frail”, “remained frail”

We modified it according to the comment (Fig 2).

Discussions

- Line 284: No citations provided to support this statement.

- Line 287: No citations provided to support this statement.

We have added a new systematic review as a citation and modified the subsequent text to match the citation (Line 273-275).

- Line 295: “No-frailty people”?

We have corrected the term you pointed out (Line 284).

- Line 312-317: A lot of this text was already included in the results section. Please make this more succinct and focused on discussion aspects.

We have removed the repetitive expressions and modified them to be more concise (Line 319-322).

- Line 337-338: Unclear what this sentence is stating.

With the re-calculation of the recovery rate from frailty, we revised the discussion regarding the change from pre-frail to robust (Line 322-325).

- Line 342: Please provide citations to support this statement.

We decided that this part was not important for the discussion and deleted it.

Conclusions 

- Again, these findings are stated quite strongly for an observational study. It should be clear that these findings are associations. 

We have made general corrections to the content of this section (Line 376-380).

---

## [Editor Report · Decision Letter 2]

5 Feb 2021

Social factors associated with reversing frailty progression in community-dwelling late-stage elderly people: An observational study

PONE-D-20-26897R2

Dear Dr. Takatori,

We’re pleased to inform you that your manuscript has been judged scientifically suitable for publication and will be formally accepted for publication once it meets all outstanding technical requirements.

Kind regards,

Simone Reppermund, PhD

Academic Editor

PLOS ONE
---

## [Editor Report · Acceptance letter]

9 Feb 2021

PONE-D-20-26897R2 

Social factors associated with reversing frailty progression in community-dwelling late-stage elderly people: An observational study 

Dear Dr. Takatori:

I'm pleased to inform you that your manuscript has been deemed suitable for publication in PLOS ONE. Congratulations! Your manuscript is now with our production department. 

Kind regards, 

on behalf of

Dr. Simone Reppermund 

Academic Editor

PLOS ONE